# A Comprehensive Assessment of Ultraviolet-Radiation-Induced Mutations in *Flammulina filiformis* Using Whole-Genome Resequencing

**DOI:** 10.3390/jof10030228

**Published:** 2024-03-20

**Authors:** Qianhui Huang, Xing Han, Zongjun Tong, Youjin Deng, Luyu Xie, Shengrong Liu, Baogui Xie, Weirui Zhang

**Affiliations:** 1Mycological Research Center, Fujian Agriculture and Forestry University, Fuzhou 350002, China; 2College of Life Science, Ningde Normal University, Ningde 352100, China; elberthuang123@163.com (Q.H.);; 3Fujian Higher Education Research Center for Local Biological Resources, Ningde 352100, China; 4College of Computer Science, National University of Singapore, Kent Ridge, Singapore 117417, Singapore; 5Institute of Urban Agriculture, Chinese Academy of Agricultural Sciences, Chengdu 610000, China; hanxing@caas.cn (X.H.); ttzjun1@163.com (Z.T.)

**Keywords:** UV radiation, enoki, base substitution, tandem mutation, fragment deletion

## Abstract

Nucleotide substitutions have played an important role in molecular evolution, and understanding their dynamics would contribute to genetic studies. Related research with defined DNA sequences lasted for decades until whole-genome sequencing arose. UV radiation (UVR) can generate base changes and other genetic variations in a short period of time, so it would be more meaningful to explore mutations caused by UVR from a genomic perspective. The monokaryon enoki strain WT583 was selected as the experimental material in this study because it can spontaneously produce large amounts of oidia on PDA plates, and the monokaryons originating from oidia have the same genotype as their mother monokaryon. After exposure to UV radiation, 100 randomly selected mutants, with WT583 as the reference genome, were sent for genome sequencing. BWA, samtools, and GATK software were employed for SNP calling, and the R package CMplot was used to visualize the distribution of the SNPs on the contigs of the reference genome. Furthermore, a k-mer-based method was used to detect DNA fragment deletion. Moreover, the non-synonymous genes were functionally annotated. A total of 3707 single-base substitutions and 228 tandem mutations were analyzed. The immediate adjacent bases showed different effects on the mutation frequencies of adenine and cytosine. For adenine, the overall effects of the immediate 5′-side and 3′-side bases were T > A > C > G and A > T > G > C, respectively; for cytosine, the overall effects of the immediate 5′-side and 3′-side bases were T > C > A > G and C > T > A > G, respectively. Regarding tandem mutations, the mutation frequencies of double-transition, double-transversion, 3′-side transition, and 5′-side transition were 131, 8, 72, and 17, respectively. Transitions at the 3′-side with a high mutation frequency shared a common feature, where they held transversions at the 5′-side of A→T or T→A without covalent bond changes, suggesting that the sequence context of tandem motifs might be related to their mutation frequency. In total, 3707 mutation sites were non-randomly distributed on the contigs of the reference genome. In addition, pyrimidines at the 3′-side of adenine promoted its transversion frequency, and UVR generated DNA fragment deletions over 200 bp with a low frequency in the enoki genome. The functional annotation of the genes with non-synonymous mutation indicated that UVR could produce abundant mutations in a short period of time.

## 1. Introduction

Nucleotide substitutions are pivotal in promoting molecular evolution, and understanding their dynamics would contribute to genetic studies, such as on phylogenic reconstruction [1], the estimation of nucleotide substitution bias [2], and the relationship between base substitutions and some diseases [3,4]. Spontaneous mutations occur slowly and cause genome changes via single-base substitutions, insertions, deletions, and duplications [5,6], while mutagen-induced mutations occur explosively in a short period, resulting in similar mutation spectra to spontaneous mutations [7,8,9,10]. 

Previous studies have extensively investigated the mutational effects of ultraviolet radiation (UVR), the most pervasive mutagen, on DNA. Solar radiation is the predominant causal factor for squamous cell carcinoma [11]. UVR mainly produces DNA photoproducts such as cyclobutane pyrimidine dimer (CPD) and pyrimidine (6-4)pyrimidone (64PP) [12], and some other photoproducts in lower amounts, such as Dewar valence isomers and 8-Oxo-7,8-dihydroguanine [13]. Base changes occur after these photoproducts are incorrectly repaired. UVR has specific mutation spectra, the so-called UV signature. According to wavelength, UVR is divided into ultraviolet A (UVA; wavelength: 320–400 nm), ultraviolet B (UVB; wavelength: 290–320 nm), and ultraviolet C (UVC; wavelength: 100–290 nm) [14]. Although UVA, UVB, and UVC can produce UV signature mutations, such as C-to-T transition and CC-to-TT tandem transition, they show differentiated mutagenicity on DNA [15,16,17,18]. The mutation spectra of UVB are more similar to those of UVC than UVA [18], probably because the energy absorption of UVA, UVB, and UVC in DNA is different in terms of wavelength. The yields of CPD and 64PP are highest at approximately 260 nm, which is in the action spectra of UVC and parallel with the absorption spectra of DNA [12]. UVA radiation cannot be sufficiently absorbed compared with UVB [4], thus contributing to the differential mutagenicity between UVA and UVB. It is widely accepted that UVB and UVC are the main producers of CPD and 64PP and that UVA mainly causes mutations by producing reactive oxygen species [4,19,20,21]. In addition to base substitutions, deletions are the sparsest and least studied mutations induced by UVR. In earlier studies, researchers have explored deletions caused by double-strand breaks using single linear DNA fragments with defined sequences [22,23]; therefore, deletion-related studies have been limited. 

Different protocols have been used to detect UV-induced mutations. In early studies, irradiated DNA was often treated with polymerase-exonuclease, T4 endonuclease V, piperidine, or dimer-specific endonuclease to estimate photoproduct distributions [24,25,26]. As PCR technology spread after the 1980s, more sequence-defined DNA fragments [27,28,29,30] were sequenced to detect UV-induced genetic changes. Whole-genome sequencing technology has promoted the discovery of new and specific mutations, such as genomic-level and cancer-specific mutations [31,32]. Detecting mutations at the genomic level would unravel new mechanisms underlying specific phenotypes induced by UV.

Limited by technology or cost, previous UV-induced mutational experiments were performed with defined DNA sequences or in specific models [25,26,30], which might have resulted in narrow mutational spectra and could not present a systematic perspective. From this viewpoint, we chose the oidia of *Flammulina filiformis* as the material for mutational experiments, expecting wide mutational spectra. *F. filiformis* can spontaneously produce large amounts of oidia when cultivated on a PDA Petri dish, and the oidia have the same genotype as their mother monokaryons, with a genome of approximately 37 Mb in size. We mutated the oidia of the monokaryon *F. filiformis* strain WT583 using UV light. After recovery, 100 randomly selected mutants and WT583 were sent for whole-genome sequencing. We precisely detected and located base changes in the WT583 genome and analyzed two typical UV signature mutations. We also characterized the distribution of point mutations and found DNA fragment deletions in the genomes of mutants. Besides mutational experiments on human skin cancer cells and *Escherichia coli* [31,32], our research is among the few studies on UV-induced mutations and whole-genome sequencing.

## 2. Materials and Methods

### 2.1. Oidium Collection

The monokaryon strain WT583 of *F. filiformis* used in this study originated from the protoplast monokaryogenesis of ′Nongjin 6′, which was bred and stored in the Mycological Research Center at Fujian Agriculture and Forestry University. The mycelia of WT583 are prone to producing large amounts of oidia on potato dextrose agar medium (1 L of medium, comprising 200 g of potato, 20 g of dextrose, and 20 g of agar, sterilized at 121 °C for 30 min). The oidium collection was carried out according to Kemp [33], with some modifications. WT583 was inoculated on PDA plates covered with autoclaved cellophane. When the plates were fully covered by mycelia, the cellophane with mycelia was put into a 50 ml sterile tube with 20 ml of sterile water. After shaking the tube, the oidium suspension was filtered via a funnel with a fritted disc. The number of oidia in the suspension was estimated using a hemocytometer and adjusted to 1000 oidia per milliliter. An amount of 100 µL of the oidium suspension was evenly spread on PDA plates and incubated at 24 °C for one week. Then, the germination rate was calculated according to the number of visible colonies on the 10 plates, which laid the basis for calculating the lethality rate.

### 2.2. Mutation Experiment

UV radiation was applied on a clean bench with an 8 W UV lamp (manufacturer: Philips; wavelength: 254 nm). UVC was the sole component of the UVR in our experiments. A magnetic stirrer was used, and the rotating speed was set at 400 r/min to ensure the homogeneousness of the mutation. The stirrer was positioned vertically, 22.5 cm from the UV lamp. An amount of 5 ml of the diluted oidia suspension was poured onto a 9 cm plate for irradiation. Irradiation lasted from 0 to 45 s with a 5 s gradient. Then, 100 µL of the radiated suspension was evenly spread onto PDA plates and incubated at 24 °C for one week, and the lethality rate was calculated incorporating the germination rate. All these operations were performed in darkness. Mutants that recovered at a 90% lethality rate were cultivated on PDA plates. One hundred randomly selected mutants, together with WT583, were sent for whole-genome resequencing. The lethality rate and part of the mutant colonies were uploaded as Appendix A. One biological replicate was performed.

### 2.3. Whole-Genome Resequencing

The genomic DNA of all mutants was prepared from mycelia using a modified CTAB DNA extraction protocol (see Appendix A) [34]. All genomic DNA specimens were submitted to Novogene Co., LTD (Beijing, China). Agarose gel electrophoresis was employed to analyze the purity and integrity of the genomic DNA, and Qubit 2.0 (Invitrogen, Carlsbad, CA, USA) was used to precisely determine the density. After fragmentation with a Covaris sonicator (Covaris, Woburn, MA, USA), 101 DNA libraries were constructed and sequenced on the Illumina HiSeq X Ten platform at a 100× sequencing depth. FastQC (http://www.bioinformatics.babraham.ac.uk/projects/fastqc, 4 October 2018) was used to remove low-quality reads or reads with adapter or poly-N. Finally, approximately 3 Gb 150-bp paired-end reads with a high quality (Q20 > 90% and Q30 > 85%) were obtained from each library. 

### 2.4. Base Change Calling and Distributions of Single-Base Mutations on Contigs

Clean reads of all mutants were mapped onto the genome of WT583 as the reference genome. Initial alignments were performed with BWA v 0.7.17 [35] combined with samtools v 1.7 [36]. Subsequent realignments and SNP discoveries were carried out with GATK [37], based on the Variant Filtration module. All software parameters were set according to Zhu et al. [38]. The filter conditions were QD < 2.0, MQ < 40.0, FS > 60.0, SOR > 3.0, MQRankSum ≤ 12.5, and ReadPosRankSum ≤ 8.0. SNPs meeting any of the conditions were eliminated. BAM files from alignments were visualized with the IGV software with the default parameter settings to filter false-positive SNPs [39]. All single-base substitutions are listed in Appendix A. R package CMplot (https://github.com/YinLiLin/R-CMplot, 20 January 2019) was employed to visualize the positions of all the eligible mutation sites on the contigs of the WT583 genome according to its introduction. 

### 2.5. Effects of Immediately Adjacent Bases on Mutation Frequency and Tandem Mutation

The total numbers (Ns) of motifs 5′-AA-3′, 5′-TA-3′, 5′-CA-3′, 5′-GA-3′, 5′-AT-3′, 5′-AC-3′,5 ′-AG-3′, 5′-AC-3′, 5′-TC-3′, 5′-CC-3′, 5′-GC-3′,5 ′-CA-3′, 5′-CT-3′, and 5′-CG-3′ were collected from the genome of *F. filiformis* (the corresponding numbers are listed in Appendix A) to analyze the effect of immediately adjacent bases on base substitutions. The standardized mutation frequency was obtained by dividing the real numbers of mutated motifs by their total numbers and then multiplying by 10^6^. Based on Appendix A, two adjacent bases in one mutant were extracted from the genome to analyze tandem mutations. 

### 2.6. Detection of DNA Fragment Deletion

DNA fragment deletions were detected using WEI’s method [40], whereby deletions of over 200 base pairs were sensitively detected and returned by the server after filtering deletions of less than 200 bp. The working principle of deletion detection is as follows: Clean reads of the mutants and WT583 from genome resequencing were mapped onto each other. Then, the differentiated sequences were processed and pooled as two k-mer libraries, i.e., one library exclusively contained sequences from WT583, and the other library only contained sequences from the mutants. The alignments of the k-mer libraries with the reference genome were visualized using IGV software with the default parameters. WEI′s research group developed an online platform on Aliyun (http://www.mrclab.top/, 20 February 2019) on which SNP calling and deletion detection can be conducted. The primers for PCR detection were designed with primer v5.0 according to the returned sequences (Premier Biosoft International, Palo Alto, CA, USA). The two primer pairs designed in this study, alongside the PCR program and PCR components, are listed in Appendix A.

### 2.7. Functional Annotation of Mutated Genes

Firstly, GeneMark-ES (version 2) [41] was employed to establish the gene model based on the genome of WT583 and analyze the genomic distribution of all the mutation sites. ANNOVAR [42] was utilized to annotate the VCF file generated with VarScan [43] according to alignments between the mutants and WT583. Finally, the genetic distributions of all the mutation sites were extracted and calculated. We utilized GeneMark-ES because it uses a new algorithm that can improve prediction accuracy in fungal genomes [44]. Secondly, all the mutated genes were extracted for functional annotation using the online software eggNOG 5.0 with the default parameters [45]. Consequently, the OMICSHARE platform (http://www.omicshare.com/tools, 20 February 2019) was employed to perform GO and KEGG annotations according to the GO and KEGG numbers returned from eggNOG 5.0 (http://eggnog6.embl.de/#/app/home, 20 February 2019).

## 3. Results

### 3.1. Single-Base Substitutions and Their Distributions on Contigs

A summary of the quality of next-generation sequencing is listed in Table 1, and detailed information on the sequencing quality is shown in Appendix A. The sequencing quality of all genomes was sufficient for subsequent analysis. After eliminating false-positive mutation sites, 3707 mutation sites were obtained with high confidence and good quality. The transition frequency was much higher than the transversion frequency (Table 2). Regardless of the strand specificity, all the base substitutions were treated as the transition and transversion of adenine and cytosine. There were 1180 sites with mutated adenine, with a transition frequency of 55.17%, and 2527 sites with mutated cytosine, with a transition frequency of 88.60%, suggesting cytosine was apt to mutate, and its transition frequency was much higher than that of adenine. 

UV-induced mutations were unevenly distributed in the genome of WT583 (Figure 1 and Table 3; for related data, see Appendix A). Nine short contigs—tig14, tig15, tig18, tig41, tig56, tig109, tig110, tig111, and tig112—spanning 23 566 bp to 678 856 bp, did not hold any mutation sites (Table 3). Eleven other short contigs—tig25, tig30, tig48, tig52, tig54, tig106, tig108, tig113, tig115, tig118, and tig124—contained 1 to 55 mutation sites (Table 3). The remaining 12 longer contigs bore most of the mutation sites, and the average distance between two mutation sites on these contigs ranged from 8 502 bp to 18 307 bp (Table 3). Three contigs—tig20, tig46, and tig121—contained some long regions without mutation sites (Figure 1).

### 3.2. Effects of Immediately Adjacent Bases on Mutation Frequency

We extracted all 3707 three-base motifs with an intermediate mutated base to assess the effects of the immediately adjacent bases on the mutation frequencies of adenine and cytosine. The detailed three-base motifs are listed in Appendix A. For base substitutions of adenine, the overall effects of bases at the 5′-and 3′-sides were T > A > C > G and A > T > G > C, respectively (Figure 2A,B). For base substitutions of cytosine, the overall effects of bases at 5′-and 3′-sides were T > C > A > G and C > T > A > G, respectively (Figure 2C,D). The four bases at the 5′-side of adenine showed the same promoting effects on each type of base substitution with their overall effects, i.e., T > A > C > G. In addition, adenine at the 3′-side and thymine at the 5′-side promoted the transition of adenine to guanine (Figure 2A,B). Moreover, cytosine at the 3′-side and thymine at the 5′-side elevated the transition frequency of cytosine (Figure 2C,D).

We also observed a discrepancy wherein the transition frequency was not always higher than the transversion frequency (Figure 2B). Pyrimidines at the 3′-side of adenine, i.e., thymine and cytosine, could be conducive to its transversion to thymine. A chi-square test was performed to clarify the effect of pyrimidine at the 3′-side on adenine transversion. The results positively supported the promoting effect of pyrimidine on adenine transversion (Table 4).

### 3.3. Tandem Mutations

Ten kinds of tandem mutations occurred at 228 sites in the mutant genomes (Figure 3A–J; for detailed data, see Appendix A). To assess the effects of tandem-base sequences on tandem mutations, tandem mutations were divided into four types, i.e., double-transition (transition occurs at both bases), double-transversion (transversion occurs at both bases), transition at the 3′-side (transition occurs only at the 3′-side of the two bases), and transition at the 5′-side (transition occurs only at the 5′-side of the two bases). The mutation frequencies of double-transition, double-transversion, transition at the 3′-side, and transition at the 5′-side were 131, 8, 72, and 17, respectively (Figure 3K). We observed that transitions at the 3′-side with a high mutation frequency shared a common feature, where they exhibited transversions at the 5′-side A→T or T→A, without covalent bond changes. Meanwhile, for those with a low mutation frequency, transversions at the 5′-side produced covalent bond changes, such as A→C, C→A, G→T, and T→G. Therefore, double-transitions and transitions at the 3′-side without covalent bond changes could occur more frequently than other tandem mutations.

### 3.4. DNA Fragment Deletions

Two deleted DNA fragments spanning approximately 720 bp and 290 bp (Figure 4A,B) were detected with software and verified via PCR (Figure 4C). The alignment showed that both the deleted fragments existed in the genome of WT583 but not those of the mutants. No bands were visible when using the genomic DNA of the mutants as the template; however, solid bands were visible when using the genomic DNA of WT583 as the template, indicating no corresponding sequences in the mutant genomes. The amplified bands were larger than those output from the software because the primers used were in the flanking regions of the deleted regions.

### 3.5. Annotation of Genes with Non-Synonymous Mutations

The distribution of mutation sites in intergenic, intron, exon, and core promoter regions was characterized (Table 5). Among the 1458 mutation sites on exons (for information on genes with non-synonymous mutations, see Appendix A), 617 genes with non-synonymous mutations were annotated as 33 GO items included in Biological Process, Cellular Component, and Molecular Function (Figure 5A). In Biological Process, genes with non-synonymous mutations were mostly annotated as cellular process, metabolic process, and single-organism process. In Cellular Component, genes with non-synonymous mutations were mainly annotated as membrane, cell, cell part, membrane part, and organelle. In Molecular Function, genes with non-synonymous mutations were mainly annotated as catalytic activity and binding. In total, 430 genes with non-synonymous mutations received KEGG annotations (Figure 5B). A total of 37, 31, 29, 27, 25, 24, and 20 genes with non-synonymous mutations fell into KEGG A-class catalogs, i.e., carbohydrate metabolism; signal transduction; translation; amino acid metabolism; folding, sorting, and degradation; cell growth and death; and transcription, respectively. We also investigated KEGG B-class items based on A catalogs (Table 6). These B-class items, such as amino acid metabolism, the Ras signaling pathway, and the MAPK signaling pathway, are vital in many biological processes. These data indicate that UVR can generate abundant genetic mutations in a short period of time.

## 4. Discussion 

In this study, we investigated typical UVR-induced mutations based on whole-genome sequencing. Transitions (C:G→T:A and A:T→G:C) occurred at the highest frequency, with C:G→T:A transitions accounting for 60.40%. This result supports previous studies from a genomics perspective. UV-induced C:G→T:A transitions always accounted for the highest proportion of total substitutions in both eukaryotic and prokaryotic cells [30,46,47,48]. UVR mainly produces the photoproducts CPD and (6-4)PP in DNA [49,50]. Then, CPD and (6-4)PP are repaired by polymerases, resulting in C:G→T:A transitions [29,51], which explains why C:G→T:A transitions are the predominant component of UV-induced mutations. Previous studies on UV-induced mutations primarily focused on C:G→T:A transitions, while very few groups reported on A:T→G:C transitions, which ranked second among the six mutation types. Kamiva et al. found that A:T→G:C transitions mainly resulted from the photoproduct T(6-4)T with adjacent adenines in both single-strand DNA and double-strand DNA [52], corresponding to our results. Moreover, mutations are easier at the T(6-4)T site than at the T = T dimer site [52,53]. Thus, it would be meaningful to study A:T→G:C transitions.

Both the immediately adjacent bases significantly influenced the base substitutions of the intermediate bases, while both the following bases showed much less influence [54]. Thus, we focused on the effect of both the immediately adjacent bases. In a review of the previous literature, we found that researchers focused on the effects of bases at the 5′-side on the mutated bases [55,56], for they thought the effects of bases at the 3′-side were not significant. In our study, adenine at the 3′-side of adenine contributed more to adenine’s transition frequency than adenine at the 5′-side, which was also true for guanine at the 3′-side of adenine, and adenine, cytosine, and guanine at the 3′-side of cytosine. Therefore, we proposed that bases at the 3′-side would also significantly contribute to the transition frequency or the overall mutation frequency. A previous study obtained similar results to ours in bacterial genomes [31]. Cytosine and thymine at the 3′-side might be conducive to the discrepancy we observed because cytosine and thymine belong to pyrimidine. This needs further evidence and discussion. Bases at the 3′-side noticeably affected the mutation frequency of the immediately adjacent bases. Future research should focus on the effect of bases at the 3′-side on base substitutions. The immediately adjacent bases of cytosine affected its transition—T > C > A > G and C > T > A > G, respectively—indicating that pyrimidines contributed more to the cytosine transition. In human skin cancer cells, UV-induced mutations featured C:G→T:A transitions at 5’-TCG-3’ [4], which showed that the 5’-TC-3’ motif can result in a high cytosine transition frequency. Lee and Pfeifer’s study showed that the 5’-TCG-3’ and 5’-CCG-3’ motifs resulted in a high mutation frequency under UVR [57]. Both studies showed that the 5’-TC-3’ motif increased the mutation frequency. In addition, our study showed that motifs 5’-TCT-3’ and 5’-TCC-3’ exhibited the highest mutation frequencies (352 and 348, respectively), while motifs 5’-TCG-3’ and 5’-CCG-3’ exhibited only 216 and 92 mutation sites (see Appendix A), respectively. We ascribed the difference between their and our studies to the different DNA sequences, for they employed defined sequences or vectors with methylated or phosphorylated bases. Although different results were obtained by using altered DNA materials, it is widely accepted that base substitutions are significantly affected by the neighboring nucleotide composition [58,59]. The mutation-promoting influence of thymine on its nearest base may be because the A and T composition could decrease the repairing efficiency of DNA and stabilize the mismatch between guanine and adenine [60,61].

Limited by technology, earlier research often qualitatively visualized the distribution of UV-induced mutations on short and sequence-defined DNA fragments [26,27]. Next-generation sequencing technology and bioinformatics promote the visualization of mutation distributions in whole genomes. Hodis et al. applied whole-exome sequencing technology to quantitatively reveal mutation distributions in melanoma [62]. Although they employed different methods, their results indicated non-random distributions in defined DNA fragments as well as whole genomes. Some factors affect the distribution of point mutations induced by UVR. Nucleosomes modulate the mutation distribution of both CPD and (6-4)PP. The photoproducts in DNA wrapped in nucleosomes showed a similar distribution pattern to naked DNA [63]. Nucleosomes differentially influence the distributions of CPD and (6-4)PP. The distribution of (6-4)PP dimers is more random than that of CPDs or the total UV photoproducts, indicating that histones have different roles in the formation of CPD and (6-4)PP [24]. The distribution of CPDs in the genome of cutaneous melanomas exhibits a periodical oscillation along with the periodical existence of histone octamers [64]. The formation of CPD and (6-4)PP in specific base contexts also affects the distribution of mutations induced by UV light. In skin cancer cells, CPDs form at the PyrmCpG motif [65], while (6-4)PP forms at 5′-TpC and 5′-CpC [28]. These motifs are non-randomly distributed in DNA [65]; therefore, these factors may contribute to the non-random distribution of mutations induced by UVR.

We observed another UV signature mutation, CC→TT, a typical tandem-base mutation. This well-known mutation in skin cells was generated by sunlight and frequently occurred at tandem pyrimidine sites [66,67,68]. We also observed another tandem-base mutation, GC→AT, with a high mutation frequency, where both bases occurred as a transition. Previous studies showed that cytosine was prone to forming CPD with its 5′-side pyrimidine bases and, subsequently, CPD deaminated, resulting in a C→T transition with a mutation frequency reaching 80% [4,69]. Thus, CC→TT tandem mutations may result from double-deamination at CC sites. However, double-deamination may not explain why GC→AT occurred at a high frequency. Tandem mutations with 3′-side transitions occurred frequently; however, only the portion with a 5′-side T→A or A→T transversion held a higher mutation frequency. Compared with the portion with a low mutation frequency, a chemical bond number change at the 5′-side of the tandem base may affect the mutation frequency of such a tandem mutation. However, the underlying mechanism of this less-known tandem mutation type needs further investigation. 

UV-induced DNA fragment deletions belong to structural variations (SVs), which include insertions, duplications, and copy number variations [70]. SVs in the literature often exclusively refer to alterations of DNA fragments spanning from 50 bp to metabases [21]. However, deletions under 200 bp were not detected and returned by the server in our studies, resulting in the incomplete detection of SV spectra. The deleted fragments caused by double-strand breaks usually have two different kinds of destinies, i.e., some are inserted back into the chromosome, while others drift away from the nucleus and are digested by nucleases [71]. Our study showed that the deleted fragments were not inserted into the chromosome, as their corresponding k-mers were not visualized in the IGV window. Therefore, we proposed that both the deleted fragments might have been digested in the cytoplasm. 

In conclusion, the genomes of 100 randomly selected mutants generated via UVR were sequenced. Subsequent analysis mainly focused on two kinds of UV signatures, i.e., single-base substitutions and tandem mutations. The immediately adjacent bases differentially affected the mutation frequencies of adenine and cytosine. Bases at the 3′-side also significantly influenced the transition frequency. Unlike the noticeable motif 5′-TCG-3′ in cancer induced by UV, motifs 5′-TCT-3′ and 5′-TCC-3′ in our studies exhibited the highest mutation frequencies. The sequence contexts of the tandem mutations affected the mutation frequency. The locations of all the 3707 single-base substitutions revealed a non-random distribution. Deletions, another rare type of UVR-induced mutation, were detected in only two mutants. The functional annotations of the non-synonymous genes indicated that UVR could be conducive to the mutational abundance in enoki mushrooms in a short period, which causes abundant phenotypic variations. Thus, UVR would be the best choice to produce mutants with visible phenotypic changes, or for other genetic studies, combined with its convenience and low cost.

## Figures and Tables

**Figure 1 jof-10-00228-f001:**
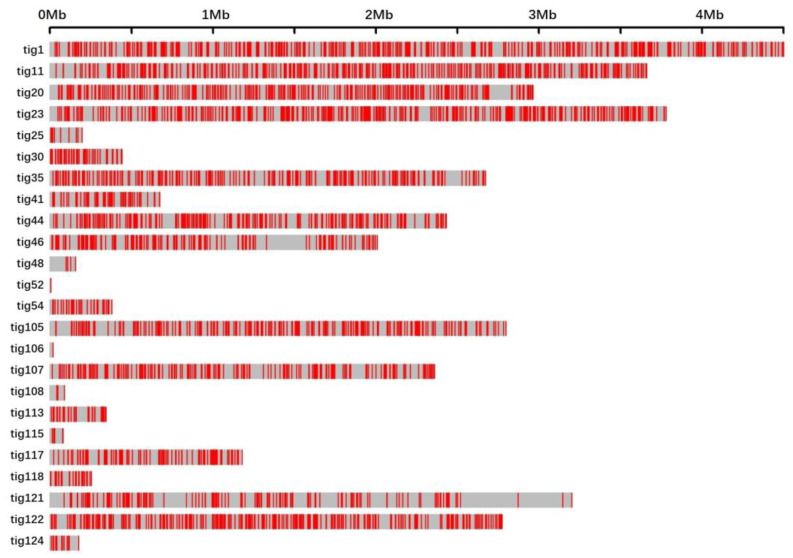
Distribution of mutated sites on all the contigs of WT583 genome. The upright red bars represent mutation sites.

**Figure 2 jof-10-00228-f002:**
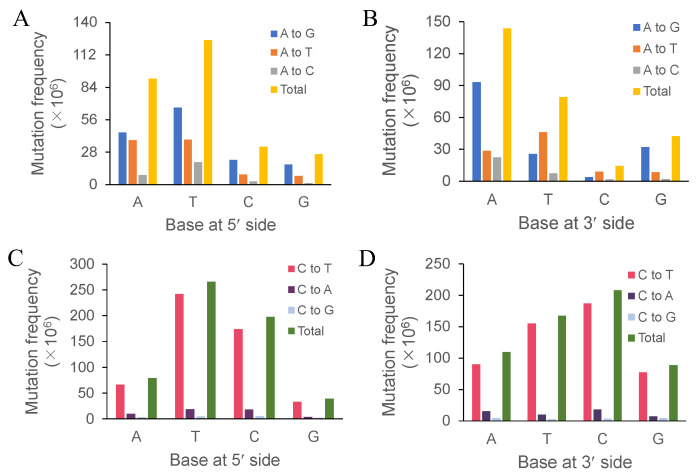
Effects of flanking bases on base substitutions of adenine and cytosine. The mutation frequencies of adenine (**A**,**B**) and cytosine (**C**,**D**) under the effects of flanking bases at both the 3′- and 5′-sides were calculated. A, T, C, and G on the X-axis represent adenine, thymine, cytosine, and guanine, respectively. “A to G”, “A to T”, and “A to C”, and “C to T”, “C to A”, and “C to G” indicate the transition and transversion types of adenine and cytosine, respectively. The number (N) of each motif in the whole genome of *F. filiformis*, i.e., 5′-AA-3′, 5′-TA-3′, 5′-CA-3′, 5′-GA-3′, 5′-AT-3′, 5′-AC-3′, 5′-AG-3′, 5′-AC-3′, 5′-TC-3′, 5′-CC-3′, 5′-GC-3′, 5′-CA-3′, 5′-CT-3′, and 5′-CG-3′, was calculated. The number (n) of each mutated motif was divided by N, and the dividend multiplied by 10^6^ generated the final standardized mutation frequency.

**Figure 3 jof-10-00228-f003:**
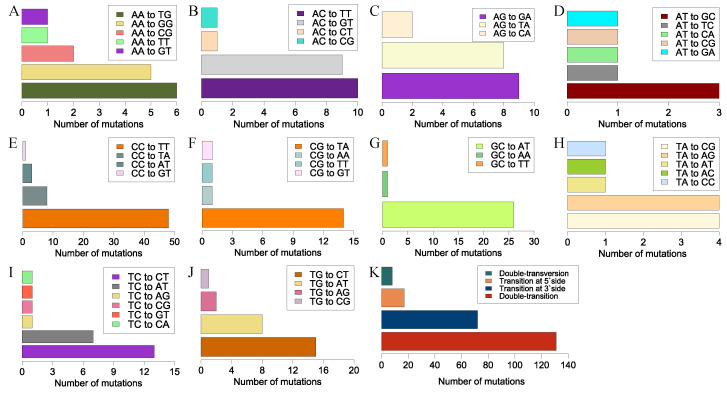
UV-induced tandem mutations in mutant genomes. These tandem motifs came from one strand without considering strand complementarity. Subfigures (**A**–**J**) show the mutation numbers of tandem motifs. Subfigure (**K**) summarizes all tandem mutations according to substitution types of the mutated motifs, i.e., double-transition (transition occurs at both bases), double-transversion (transversion occurs at both bases), transition at the 3′-side (transition occurs only at the 3′-side of the two bases), and transition at the 5′-side (transition occurs only at the 5′-side of the two bases).

**Figure 4 jof-10-00228-f004:**
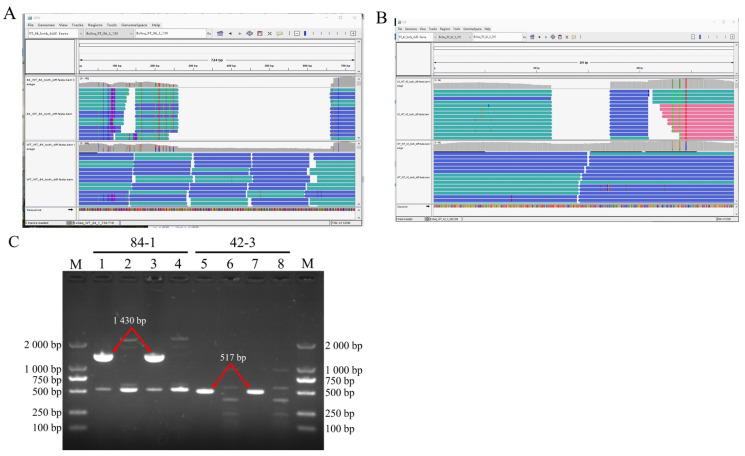
Deleted DNA fragment detection. Two deleted fragments, spanning approximately 720 bp (**A**) and 290 bp (**B**), were visualized with IGV software after mapping onto the reference genome. The two fragments were named 84-1 and 42-3 according to the IGV output. PCR was performed to verify the results (**C**). Channels 1-4 were for 84-1, and channels 1 and 3 were used as controls for channels 2 and 4. Channels 5-8 were for 42-3, and channels 5 and 7 were used as controls for channels 6 and 8. The genomic DNA of WT583 was used as the template in the control experiment.

**Figure 5 jof-10-00228-f005:**
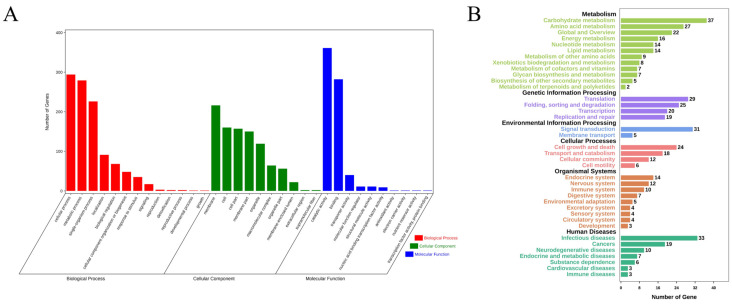
Functional annotations of genes with non-synonymous mutations. GO analysis (**A**) includes three main items with different colors, i.e., Biological Process, Cellular Component, and Molecular Function. The Y-axis represents the number of genes with non-synonymous mutations. KEGG analysis (**B**) contains six A catalogs, i.e., Metabolism, Genetic Information Processing, Environmental Information Processing, Cellular Processes, Organismal Systems, and Human Diseases. Here, Human Diseases means that these mutated genes targeted homologous genes in the human genome related to diseases. The horizontal X-axis represents the number of genes with non-synonymous mutations.

**Table 1 jof-10-00228-t001:** Quality index of whole-genome sequencing.

Quality Index	Average	Minimum
Seq_depth (✕)	65.39	35.93
Q20 (%)	96.07	95.02
Q30 (%)	90.73	88.71
GC content (%)	48.31	42.62

Note: Next-generation genome sequencing quality of 100 mutants and WT583.

**Table 2 jof-10-00228-t002:** Proportion of different base change types.

Substitution Types	No. of Mutation Sites	Ratio
G:C→A:T	2239	60.40%
A:T→G:C	651	17.56%
A:T→T:A	394	10.63%
G:C→T:A	221	5.96%
A:T→C:G	135	3.64%
G:C→C:G	67	1.81%
Total	3707	100%

Note: Base substitution contains transition and transvertion. G:C→A:T and A:T→G:C belong to transition, A:T→T:A, G:C→T:A, A:T→C:G, and G:C→C:G belong to tansversion.

**Table 3 jof-10-00228-t003:** Number of mutated sites on all contigs.

Contig	Mutated Sites per Contig	Contig Length (bp)	Average Length between Mutated Sites
tig1	473	4,475,816	9463
tig11	427	3,630,557	8502
tig14	0	48,161	-
tig15	0	62,023	-
tig18	0	36,591	-
tig20	316	3,033,355	9599
tig23	425	3,725,332	8765
tig25	14	222,816	15,915
tig30	55	443,364	8061
tig35	262	2,648,354	10,108
tig41	0	678,856	-
tig44	237	2,393,417	10,099
tig46	175	2,040,433	11,660
tig48	5	155,795	31,159
tig52	1	48,353	48,353
tig54	38	386,978	10,184
tig56	0	119,340	-
tig105	269	2,754,843	10,241
tig106	1	37,686	37,686
tig107	220	2,350,230	10,683
tig108	3	147,222	49,074
tig109	0	23,566	-
tig110	0	137,946	-
tig111	0	78,587	-
tig112	0	184,367	-
tig113	40	370,450	9261
tig115	6	100,769	16,795
tig117	128	1,207,482	9433
tig118	36	276,954	7693
tig121	177	3,240,382	18,307
tig122	319	2,748,975	8617
tig124	16	222,876	13,930

Note: Contig length divided by mutated sites per contig makes average length between mutated sites.

**Table 4 jof-10-00228-t004:** *Chi-square* test for the effect of thymine and cytosine at 3′ side on adenine transversion.

Bases at 3′	Muation Types		Total	*Chi-square* Test
	A→G	A→T		
A+G	523	153	676	*p* = 0.000
T+C	128	241	369	
Total	651	394	1045	

Note: The data used here were listed in Appendix A.

**Table 5 jof-10-00228-t005:** Functional distribution of mutation sites.

Mutation Region	Mutation Frequency	Proportion
Intergenic region	627	16.91%
Intron	633	17.08%
Exon (non-synonymous)	1458	39.33%
Exon (synonymous)	707	19.07%
Core promoter	282	7.61%

Note: exon (non-synonymous) means mutations occurred in exons and resulted in amino acid changes; core promoter means promoter spanned 200 bp upstreaming the initiation codon ATG.

**Table 6 jof-10-00228-t006:** Genes with non-synonymous mutation in KEGG B class pathway.

KEGG B Class	Pathway	Pathway ID	Gene Number
Carbohydrate metabolism	Amino sugar and nucleotide sugar metabolism	ko00520	10
	Starch and sucrose metabolism	ko00500	7
	Inositol phosphate metabolism	ko00562	6
Amino acid metabolism	Valine, leucine and isoleucine degradation	ko00280	6
	Glycine, serine and threonine metablism	ko00260	4
	Tyrosine metabolism	ko00350	4
	Alamine, aspartate and glutamate metabolism	ko00250	4
Translation	Ribosome biogenesis in eukaryotes	ko03008	7
	mRNA surveillance pathway	ko03015	5
Folding, sorting and degradation	Protein processing in endoplasmic reticulum	ko04141	8
	Ubiquitin mediated proteolysis	ko04120	7
	RNA degradation	ko03018	7
Transcription	Spliceosome	ko03040	13
	Basal transcription factors	ko03022	5
Signal transduction	Ras signaling pathway	ko04014	5
	MAPK signaling pathway	ko04010	5
	Phosphatidylinositol signaling syStem	ko04070	5
	PI3K-Akt signaling pathway	ko04151	5
	Wnt signaling pathway	ko04310	5
Cell growth and death	Cell cycle - yeast	ko04111	15
	Meiosis - yeast	ko04113	11
	Cell cycle	ko04110	6

Note: KEGG B class pathways origin from KEGG A catalogues and show more details for gene functions; Gene number in the table means the number of genes with non-synonymous mutation.

## Data Availability

Data are available on request from the first author.

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
