# Peer review of "A Comprehensive Assessment of Ultraviolet-Radiation-Induced Mutations in Flammulina filiformis Using Whole-Genome Resequencing"

_jof, 2024, doi:10.3390/jof10030228_

Round 1

Reviewer 1 Report

Comments and Suggestions for Authors

Previously most studies focused on several model organisms due to laborious and costly techniques. New technologies allow biologists to spread and widen number of research objects. In the presented paper effect of ultraviolet radiation on living cells is studied using F. filiformis. This and further studies will allow to compare such effects between different organisms and to investigate evolution of cell defense against UV induced damage and repair mechanisms.

Here are mistakes that I found and some questions to the content of the paper:

46 “Ultraviolet radiation (UVR) as the most pervasive mutagen, previous studies extensively investigated its mutational effects on DNA.” should be rephrased.

51 “Base changes occur after these photoproducts being repaired.” - it’s better to mention that they were “incorrectly repaired”. 

53 “wave length” should be single word?

53 What about UV-C?

59 “In even earlier studies” – maybe just “in earlier studies”.

68 “radiated”- “ irradiated” is more suitable.

92 PDA medium recipe and reagents for it’s preparation should be presented.

98 “per miniliter”, further in the text also - 106

103 “UV radiation was performed in the clean bench with an 8 W UV lamp.” It is necessary to provide UV wavelength emitted by the lamp, lamp manufacturer, etc. Overall the description of the UV irradiation experiment setup of is not sufficient. What volume of 1000 oidia per mL suspension was irradiated? What was the diameter of the flask?

What about light after UV irradiation? Photolyases (that depend on visible light) play very important role in reparation of damage induced by UV. Are they active in F. filiformis cells? It should be specified if plates were kept in the dark after UV irradiation.

112 “Genomic DNA of all mutants was prepared from mycelia using a modified CTAB DNA extraction protocol[34].” If modifications of protocol from [34] are used, they should be precisely described.

113-114 «DNA libraries were prepared according to Illumina standard sequencing protocol.» This is not enough. Exact kit and protocol should be mentioned or cited.

114 “After testing DNA quality of the mutants…” How was DNA quality tested?

114 “ultrasonicator was used for DNA fragmentation” - should be more precise (what equipment with which parameters was used for fragmentation).

116 “Xten” Maybe “HiSeq X Ten”?

118 “After filtering reads with low quality, cleans reads were subject to further analysis.” How they were filtered. If using trimmomatic, fastp or other program it should be cited correctly.

122 “BWA (version 0.7.17) combined with samtools (version 1.7)” should be cited correctly. For most described tools and programs their authors state which paper should be cited.

124 “GATK, based on the Variant Filtration module” should be cited correctly.

140 “Now this research group develops an online platform on Aliyun(http://www.mrclab.top/), on which SNP calling and deletion detection can be conducted.” It is not clear which research group? The one mentioned above [37], or the authors of the paper?

143 “primer v5.0” should be cited correctly.

148 VarScan should be cited correctly.

Table 1 contains a word for “total” in Chinese language at the bottom. Also Chinese language  in table S1.

160 and 165 “Transition frequency was much higher than transversion frequency (table 1).” and “Conclusively, UV-induced transition frequency was much higher than transversion in mutant genomes.” – repeating sentences.

173 Why it is said that “When at 5’ side of adenine and cytosine, the four bases showed the same promoting effects on each type of base substitution of adenine and cytosine with their overall effects”, when for adenine the promoting effects diminish in one manner (T>A>C>G) and for cytosine in another (T>C>A>G)?

Promoting effects of nucleotide at 5’ or 3’ position of mutation site are analyzed. What about the different total amount of different sites in F. filiformis genome? For example, for guanine nucleotide there can be 4 variants of nucleotides at 3’ position. And we have 5’-GT-3’, 5’-GA-3’, 5’-GC-3’ and 5’-GG-3’ sites. Because I don’t have WT583 genomic sequence, I took one of F. filiformis genome assembles from NCBI database and tried to find all 5’-GA-3’ sites and 5’-TC-3’ sites which correspond to complementary 5’-GA-3’ sites in the second strand. This resulted in 2767144+2762355=5529499 such sites in whole genome. For 5’-GT-3’ and corresponding 5’-AC-3’ sites there were 2253636+2241082=4494718 such sites in whole genome. So because there are more 5’-GA-3’ sites for mutation than 5’-GT-3’ sites it is possible that they can mutate more frequently due to sheer numbers. Same problem may arise for other site variants. Could difference in site numbers in F. filiformis genome impact results presented in the paper?

An important question rises about “mutation frequency” term. What does it mean exactly and how it is calculated? In Figure 1 Y-axis is marked as “mutation frequency”. In my understanding (I tried to sum “total” values and compare them with corresponding sums from table 1) Y-axis represent number of mutation sites. Is “mutation frequency” term that is used in the paper equivalent to number of mutation sites found in the experiment? It’s better to either talk about number of mutation sites, or present it as ratio of total amount of mutation sites. Or calculate some normalized ratio which may be stated as mutation frequency (e.g. mutations per kilobase of genomic DNA).

179 “When at 3’ side of adenine, thymine and cytosine, i.e. pyrimidine, could be condusive to its transversion to thymine.” Maybe it was meant: “When at 3’ side of adenine there is thymine and cytosine, i.e. pyrimidine…”. Also “condusive” should be corrected.

212 “Considering Strand Complementarity” - capital letters?

Figure 3 and Figure 5A have bad quality. It is very difficult to read text on Figure 5A and impossible on Figure 3.

PCR results are of poor quality and are poorly presented. In lane 225 it is said that fragment 42-3 is analyzed. In 227 – fragment 42-2. In supplementary S3 (Table Deleted DNA fragments and their detection primers) there is fragment 106-1. In lane description to Figure 3 – fragment 84-1. Which fragments were analyzed? Some information about molecular marker can be found only in supplementary with unprocessed image of gel. Corresponding ladder fragments should be marked. Expected PCR fragments should also be highlighted to distinguish them from unspecific ones.

245 I am not familiar with the term “Non-Synonymous Genes”. As far as I understand this relates to genes with non-synonymous mutations in coding sequence that lead to amino acid substitutions. If it is so it’s better to talk about non-synonymous mutations in genes.

250 GO items – some are written are with capital letters, some without (257 – translation).

257 What is “Floding sorting”? Maybe folding and sorting?

I am not qualified enough to assess quality of English language, but I feel that extensive editing and grammar correction is needed.

Reviewer 2 Report

Comments and Suggestions for Authors

Qian-hui Huang and colleagues present an article on the identification of randomly generated mutations using UV radiation in the fungus Flammulina filiformis. Specifically, 100 strains were analyzed using genome wide sequencing and the frequencies, distribution and functional annotation of mutants was documented. In general, the article is comprehensibly written and contains interesting information.

However, what seems to be essentially missing in my opinion is an actual scientific question. What is the purpose of this study? I agree that not many articles have employed genome sequencing, however several groups in the last 50 years have used UV mutagenesis to select, identify and clone specific genes. Why this particular fungus was chosen and not some other well annotated model system? In my opinion, if the authors are interested in UV mutagenesis at the genomic level, then they should generate and use data from different fungi and at least perform a comparative study.

The experimental results of the actual UV mutagenesis and the survival/lethality curves etc. along with growth tests of all these mutants, or at least representative phenotypes should also be shown.

In the distribution observations there are some long genomic regions without any mutation sites. This result is very interesting. What is different in these regions?

Reviewer 3 Report

Comments and Suggestions for Authors

Overall Summary: In this manuscript, Huang et al. conducted UV based mutagenesis in Flammulina filiformis and used genome sequencing to identify the mutations occurring as a result. They conduct a bioinformatic analysis of the different patterns and locations of the mutations to be able to apply this information in a generalized manner. There are some issues with the description of the methods, types of analysis conducted and the inclusion of other information, especially about the mutant strains to be able to fully understand the patterns of mutations. Specific comments are listed below.

Specific comments:

·        Line 67-74: The introduction section should also describe and reference other model systems that have been used to conduct mutation studies. Also, describe the limitations with sequencing in terms of contig assembly and the different types of short and long sequencing which can impact the confidence level in the assessment of point mutations.

·        Method 2.2: it is unclear which type of UV light was used for mutation, UV A, B or C. This needs to be mentioned along with the strength of the UV light and the rationale behind using a particular type of UV. Further, was only one set of experiments conducted? Or were there some biological replicates? This needs to be explained and if so, some data about the deviation between the biological replicates should be presented.

·        Method 2.3: the reagents for library prep as well as their source information of location need to be added here. Also, it is unclear which Illumina instrument was used along with the type of Chip used. Further the number of reads should also be included. There is also not much information about the quality check process for DNA, any analysis done with the bioanalyzer or Tapestation should be included here.

·        Results 3.1: The quality of contig assemblies, ANI values and the genomic coverage should be included as a summary table for the mutant strains evaluated.

·        The mutation distribution (of different types of mutations) within the genome of any given mutant evaluated needs to be provided as a summary. This can be provided as a list of all the mutants evaluated and the critical class of mutations and the number of mutations within those strains should be provided as a summary table. This shows the prevalence of mutations across different strains and whether they are even or not.

·        Figure 2: it is unclear how even the distribution of the tandem mutations is across different mutants evaluated. A note about the distribution of the mutations across different strains and the top 10-25 strains with a higher number of mutations should be provided.

·        Figure 3 A& B is not clear and needs to be made larger with better resolution. Also, the strains in this figure need to be labelled legibly.

·        Figure 3C: Was only a subset of mutants evaluated for these deletions? A table summarizing the presence or absence of the deletions should be added. All the PCR results in the gel should be shown in the supplemental figures for all the mutants.

·        Table 3: Is this a summary of all contigs from all strains? If so, this needs to be mentioned. Also, since each of the genomes are assembled individually, it is unclear what the distribution of these contigs is in terms of the strains. The table and figure 4 need to have an added variable of the strains in them.

·        Figure 4 and table 5 need to clarify that the number of genes is the number of genes where mutations were recorded. The legend should be modified to clearly state that.

Round 2

Reviewer 1 Report

Comments and Suggestions for Authors

I accept all the changes in revised document and answers to my questions.

Author Response

Thank you for reviewing this manuscript

Reviewer 2 Report

Comments and Suggestions for Authors

The revised manuscript has been improved, however, the authors did not respond to any of the critisism raised by this reviewer. I have no further comments for the authors.

Author Response

Dear Editor and Revewer:

Hi

With reference to Research Article ID jof-2764475 entitled “A Comprehensive Assessment of UV-induced Mutation in Flammulina filiformis Using Whole-genome Resequencing”, we are thankful for the valuable comments and suggestion on this article.

According to comments and suggestion of Reviewer 1, following corrections have been made in jof-2764475 manuscript.

Explanations made according to Reviewer 2 comments:

Qian-hui Huang and colleagues present an article on the identification of randomly generated mutations using UV radiation in the fungus Flammulina filiformis. Specifically, 100 strains were analyzed using genome wide sequencing and the frequencies, distribution and functional annotation of mutants was documented. In general, the article is comprehensibly written and contains interesting information.

However, what seems to be essentially missing in my opinion is an actual scientific question. What is the purpose of this study? I agree that not many articles have employed genome sequencing, however several groups in the last 50 years have used UV mutagenesis to select, identify and clone specific genes. Why this particular fungus was chosen and not some other well annotated model system? In my opinion, if the authors are interested in UV mutagenesis at the genomic level, then they should generate and use data from different fungi and at least perform a comparative study.

Response: 1. The purpose of the this study was added in Line 78-81.

  1. The reason why we use Flammulina filiformis was presented in Line 82-85.
  2. A comparative study with other fungi is very interesting. Thanks a lot for giving

            me such good idea. We have only mushroom species in our lab, and Flammulina

            filiformis can spontaneously produce oidia than can grow into monokaryon strain.

The experimental results of the actual UV mutagenesis and the survival/lethality curves etc. along with growth tests of all these mutants, or at least representative phenotypes should also be shown.

Response: Related materials will be uploaded as supplementary materials. Seen in Line 119-120.

In the distribution observations there are some long genomic regions without any mutation sites. This result is very interesting. What is different in these regions?

Response: Good question! It will be interesting to compare regions without mutation sites and with lots of mutation sites. It is a pity we did not think of that. However, we could incorporate this question in a comparative study between species, in which there will be more regions to be analyzed.

The latest version of manuscript was the attachment.

Reviewer 3 Report

Comments and Suggestions for Authors

Authors have addressed the comments. 

Author Response

Thank you for reviewing this manuscript

Round 3

Reviewer 2 Report

Comments and Suggestions for Authors

Thank you for your response. The manuscript has been improved.

Regarding the comment on the survival/lethality curves and related growth tests, although stated in the text that those were uploaded as supplementary material S6, however, no Figure 6 can be found in the supplementary material. Please upload the relevant file.